# Disruption of Cancer Metabolic SREBP1/miR-142-5p Suppresses Epithelial–Mesenchymal Transition and Stemness in Esophageal Carcinoma

**DOI:** 10.3390/cells9010007

**Published:** 2019-12-18

**Authors:** Chih-Ming Huang, Chin-Sheng Huang, Tung-Nien Hsu, Mao-Suan Huang, Iat-Hang Fong, Wei-Hwa Lee, Shao-Cheng Liu

**Affiliations:** 1Department of Otolaryngology, Taitung Mackay Memorial Hospital, Taitung City 950, Taiwan; mmh4621@gmail.com; 2Division of Oral and Maxillofacial Surgery, Department of Dentistry, Taipei Medical University—Shuang Ho Hospital, New Taipei City 235, Taiwan; 09520@s.tmu.edu.tw (C.-S.H.); 12460@s.tmu.edu.tw (T.-N.H.); 08686@s.tmu.edu.tw (M.-S.H.); 3School of Dentistry, College of Oral Medicine, Taipei Medical University, Taipei City 110, Taiwan; 4Department of Medical Research & Education, Taipei Medical University—Shuang Ho Hospital, New Taipei City 235, Taiwan; 18149@s.tmu.edu.tw; 5Department of Pathology, Taipei Medical University—Shuang Ho Hospital, New Taipei City 235, Taiwan; 6Department of Otolaryngology—Head and Neck Surgery, Tri-Service General Hospital, National Defense Medical Center, Taipei City 114, Taiwan

**Keywords:** SREBP1, esophageal cancer, miR-142-5p, epithelial-to-mesenchymal transition (EMT), fatostatin

## Abstract

Elevated activity of sterol regulatory element-binding protein 1 (SREBP1) has been implicated in the tumorigenesis of different cancer types. However, the functional roles of SREBP1 in esophageal cancer are not well appreciated. Here, we aimed to investigate the therapeutic potential of SREBP1 and associated signaling in esophageal cancer. Our initial bioinformatics analyses showed that SREBP1 expression was overexpressed in esophageal tumors and correlated with a significantly lower overall survival rate in patients. Additionally, tumor suppressor miR-142-5p was predicted to target SREBP1/ZEB1 and a lower miR-142-5p was correlated with poor prognosis. We then performed in vitro experiments and showed that overexpressing SREBP1 in OE33 cell line led to increased abilities of colony formation, migration, and invasion; the opposite was observed in SREBP1-silenced OE21cells and SREBP1-silencing was accompanied by the reduced mesenchymal markers, including vimentin (Vim) and ZEB1, while E-cadherin and tumor suppressor miR-142-5p were increased. Subsequently, we first demonstrated that both SREBP1 and ZEB1 were potential targets of miR-142-5p, followed by the examination of the regulatory circuit of miR-142-5p and SREBP1/ZEB1. We observed that increased miR-142-5p level led to the reduced tumorigenic properties, such as migration and tumor sphere formation, and both observations were accompanied by the reduction of ZEB1 and SREBP1, and increase of E-cadherin. We then explored the potential therapeutic agent targeting SREBP1-associated signaling by testing fatostatin (4-hydroxytamoxifen, an active metabolite of tamoxifen). We found that fatostatin suppressed the cell viability of OE21 and OE33 cells and tumor spheres. Interestingly, fatostatin treatment reduced CD133+ population in both OE21 and OE33 cells in concert of increased miR-142-5p level. Finally, we evaluated the efficacy of fatostatin using a xenograft mouse model. Mice treated with fatostatin showed a significantly lower tumor burden and better survival rate as compared to their control counterparts. The treatment of fatostatin resulted in the reduced staining of SREBP1, ZEB1, and Vim, while E-cadherin and miR-142-5p were increased. In summary, we showed that increased SREBP1 and reduced miR-142-5p were associated with increased tumorigenic properties of esophageal cancer cells and poor prognosis. Preclinical tests showed that suppression of SREBP1 using fatostatin led to the reduced malignant phenotype of esophageal cancer via the reduction of EMT markers and increased tumor suppressor, miR-142-5p. Further investigation is warranted for the clinical use of fatostatin for the treatment of esophageal malignancy.

## 1. Introduction

Esophageal cancer (EC) represents a common and malignant type of gastrointestinal cancer worldwide, with approximately 572,000 new cases and more than half of million deaths in 2018 [1]. Among all subtypes, approximately 90% cases are esophageal squamous cell carcinoma (ESCC) and ESCC is often diagnosed at the advanced stage [2,3]. The standard interventions for ESCC include surgery, surgery followed by chemotherapy and radiotherapy, or combination therapies [4]. However, treatment options become limited for patients with metastatic ESCC, and the five-year survival rate is estimated at only 15% [2,5]. Clinical evidence indicates that 90% of ESCC patients died from distant invasion and metastasis, and 36.8% patients show lymph node metastasis [4,5]. Thus, it is urgent to better understand the molecular mechanisms by which ESCC invasion and metastasis occur so that improved therapeutic and diagnostic agents can be developed.

Metastasis is a dynamic process where cancer cells spread from the primary site to the neighbor and/or distant tissues by acquiring a series of malignant capabilities [6,7,8]. More specifically, epithelial–mesenchymal transition (EMT) is a process by which epithelial cells lose cell–cell and cell–matrix interactions and apical–basal polarity, and acquire migratory abilities and other characteristics of mesenchymal cells, leading to the events of invasion and migration [9,10,11]. In the case of metastatic EC, malignant cells acquire their mesenchymal characteristics with reduced E-cadherin (epithelial marker) expression and increased vimentin (mesenchymal marker) [11,12,13]. It has been reported that transcription factors, such as the Snail family, execute EMT programs in normal and pathological conditions [12,13,14]. The presence of the Snai11 (snail family transcriptional repressor 1) gene is indispensable for EMT, and knockout of Slug (snail family transcriptional repressor 2, Snail2) strongly reduces invasion and metastases in EC [11,13,14,15,16]. Additionally, data showed that the increased levels of Snail, Slug, ZEB1, and so forth are instrumental for activating the metastasis machinery and predict poor prognosis in ESCC patients [13,14,17,18]. Recent studies showed that an increased expression of ZEB1(a key EMT initiator) is found in patients with esophageal cancer and is associated with distant metastasis and poor prognosis [19,20].

Functioning as transcription factors and the main regulatory elements of sterol biosynthesis and lipid metabolism, sterol regulatory element-binding proteins (SREBPs) are gaining recognition in their roles in different cellular processes, including tumorigenesis. Sterol regulatory element-binding protein 1 (SREBP1), also known as SREBF1 (sterol regulatory element-binding transcription factor 1), has recently been implicated in cancer progression and in association with clinical status and poor prognosis in different malignancies [21,22,23,24]. SREBP1-regulated fatty acid and lipid synthesis was reported in the pathogenesis of prostate cancer [25]. SREBP1 also has been shown to regulate fatty acid synthesis in immunosuppressive tumor-associated macrophages (TAMs); SREBP1 was evaluated as a potential novel target to augment the efficacy of immune checkpoint blockades and improve cancer immunotherapy [14]. Furthermore, SREBP1 was indicated to promote the metastasis of colorectal cells by activating NF-κB signaling and regulating the expression of MMP7 [26]. SREBP1 drives the cytoskeletal changes and invasion of endocrine-resistant ERα breast cancer cells by Keratin-80 upregulation [27]. However, the role and molecular mechanism of SREBP1 in progression, especially in the induction of EMT and metastasis, remain to be explored.

In this study, we first showed that SREBP1 was highly expressed in EC tissues and associated with lower overall survival and the level of tumor suppressor miR-142-5p. Subsequently, we demonstrated that SREBP1-silenced esophageal cancer cells significantly reduced tumorigenic properties of esophageal cancer cells, concertedly with the decreased EMT markers and increased miR-142-5p; the reverse was observed when SREBP1 was overexpressed. Equally important, a reduced miR-142-5p level in OE21 showed phenotypes similar to those in SREBP1-overexpressing OE33 cells and vice versa. Finally, we demonstrated that fatostatin treatment significantly suppressed tumorigenesis via downregulating SREBP1 and EMT markers, ZEB1 and vimentin, while upregulating miR-142-5p in vitro and in vivo.

## 2. Materials and Methods

### 2.1. Bioinformatics Analysis

Public EC databases were obtained and analyzed using Oncomine (https://www.oncomine.org) online tool [28,29]. The sample size of each cohort is listed in the Figure 1 legend. Clinical esophageal cancer databases were also obtained from The Cancer Genome Atlas (TCGA) databases for analyses; Gene Expression Viewer was used for comparing ESCC samples versus normal tissues (http://firebrowse.org/). SurvExpress software was used to calculate the overall survival rate of EC patients with high versus low SREPB1 expression. Finally, PITA, miRmap, and PicTar online software programs were used to predict targets for miR-142-5p.

### 2.2. Cell Culture and Transfection

Human esophageal cancer cell lines OE21 (ESCC) and OE33 (esophageal adenocarcinoma cells, EACC) were purchased from Merck, Sigma-Aldrich. Esophageal cancer cells were cultured and maintained according to the recommendations made by the vendor. In brief, both cell lines were maintained and passaged in RPMI-1640 (Gibco, Thermo Fisher Scientific, Inc., Taipei, Taiwan) and supplemented with 10% fetal bovine serum (FBS, Biological Industries, Kibbutz Beit-Haemek, Israel) and 1% compound antibiotics (Pen Strep, Gibco, Life Technologies, CA, USA) at 37 °C, 5% CO_2_.

### 2.3. Gene-Silencing Experiments

Gene-silencing experiments were performed using siRNA molecules (Cat# s129, ThermoFisher Scientifics, Taipei, Taiwan), negative control (Cat # 390843, ThermoFisher Scientifics, Taipei, Taiwan). The siRNA was transfected using Lipofectamine™2000 (ThermoFisher Scientific, Taipei, Taiwan) according to the manufacturer’s recommendations. SREBP1 overexpression experiments were carried out using plasmid containing ORF of SREBP1 (Cat # A6812, Genecopoeia, Taiwan) according to vendor’s protocols. The efficiency of silencing or overexpression was confirmed by Western blot and qRT-PCR. Fatostain (Cat # F8932) was purchased from Sigma-Aldrich, Taipei, Taiwan.

### 2.4. Colony Formation Assay

Control and/or transfected OE21 and OE33 cells esophageal cancer cells (2.5 × 10^3^) were plated in 6-well plates (Corning, NY, USA) with a base layer of 0.5% agarose gel and an upper layer of 0.35% agarose gel with RPMI, N_2_ supplement, 20 ng/mL of epidermal growth factor (EGF), and basic fibroblast growth factor (bFGF) and incubated for a week. Formed colonies were stained with 0.1% crystal violet in 20% methanol and counted. A colony is considered as a cluster of ≥50 cells.

### 2.5. Tumor Sphere Formation Assay

OE21 and OE33 cells esophageal cancer cells (5 × 10^3^/well) were plated in ultra-low-attachment six-well plates (Corning, NY, USA) with stem cell medium comprising of serum-free RPMI 1640 medium supplemented with 10 ng/mL basic fibroblast growth factor (bFGF) (Invitrogen, Grand Island, NY, USA), 1 × B27 supplement, and 20 ng/mL epidermal growth factor (EGF; Invitrogen). The stem cell medium was changed every 72 h. Spheres often are formed after 7 days. Micrographs of the spheres were taken and counted.

### 2.6. Real-Time PCR

Total RNA was extracted using Trizol reagent (Invitrogen Corp.) and reverse transcribed with a reverse transcription kit (Promega) according to the manufacturer’s protocols. Quantitative PCR was carried out using qPCR Master Mix (Promega). The primer sequences used in this study are listed below. Relative mRNA levels were calculated using the 2^−ΔΔ^Cq method and GAPDH was used as an internal control, while U6 was used for miR-142-5p. Primers for Real-Time PCR in this study are listed in Table 1.

### 2.7. SDS-PAGE and Western Blots

Total cellular protein lysates were collected and analyzed using standard SDS-PAGE and Western blotting protocols. The membranes were blocked using 10% skim milk in TBST for 30 min at room temperature, followed by the incubation with primary antibodies at 4 °C overnight. All antibodies were purchased from Cell Signaling Technology (CST, MA, USA) unless otherwise specified: SREBP1 (ab191857; 1:1,000 dilution), ZEB1 (ab228986; 1:2,000 dilution) from Abcam (Abcam, MA, USA), vimentin (#5741; 1:1000) from Cell Signaling Technology (cell signaling, Danvers, MA, USA), and E-cadherin (20874-1-AP; 1:1,000) and GAPDH (10494-1-AP; 1:10,000) from Proteintech Group (Proteintech, IL, USA), as shown in Appendix A. Membranes were then washed with TBST and incubated with the secondary antibody, horseradish peroxidase (HRP) conjugate (1:5,000), for 1 h at room temperature, washed with TBST. The immunoreactions were then carried out using an ECL kit (Amersham Biosciences, GE Healthcare, Chicago, IL, USA). GAPDH served as an internal control.

### 2.8. Flow Cytometric Analysis

OE21 and OE33 cells (parental and/or spheres) were harvested using trypsin-EDTA and washed 3 times with PBS (0.1% BSA). Subsequently, 1 × 10^6^ cells suspended in PBS (0.5% BSA) were incubated with allophycocyanin (APC)-conjugated fluorescence-labeled mouse anti-human CD133/2 (Miltenyi Biotec, Bergisch Gladbach, Germany) or APC-conjugated isotype control mouse IgG2b for 20 min at room temperature in the dark. The isotype control was added for gating. The labeled cells were sorted by flow cytometry into CD133+ or CD133− groups, and the data were collected and analyzed using the BD FACSCantoTM flow cytometry system (BD Biosciences, CA, USA).

### 2.9. In Vitro Migration and Invasion Assays

Cell invasion was assayed using 24-well Transwell plates with a pore size of 8 μm (Corning, Life Sciences, USA). For invasion assay, the Transwell inserts were precoated with 25 μL Matrigel (BD Biosciences, CA, USA) and the plates were incubated for 24 h at 37 °C, and then the transfected cells (2×10^5^) were suspended in 200 μL serum-free RPMI-1640 medium and seeded on the upper chamber. The lower chamber was filled with 600 μL RPMI-1640 containing 10% FBS as an attractant. After incubation for 24 h, the nonmigrated cells in the upper chamber were removed by cotton swab. Migrated cells were fixed with 4% cold paraformaldehyde and stained with 0.2% crystal violet (CoWin Biotech Co., Ltd., Beijing, China) for 15 min, and the stained cells were counted at 100× magnification under an inverted microscope (Olympus, Tokyo, Japan). Relative invasion was carried out at least three different experiments and presented as the average number of stained cells from five randomly chosen fields.

### 2.10. Animal Experiments

Nude mice (aged 6 weeks, 18–20 g) were purchased from BioLASCO Taiwan Co., Ltd. (Taipei, Taiwan) and maintained under specific pathogen-free conditions in the Laboratory Animal Center (LAC). After one week of acclimatization, the mice were subcutaneously injected in the right flank with 2 × 10^6^ OE21 cells (in 100 μL of PBS). All mice were then randomly divided into the vehicle control (saline, i.p. injection) and fatostain treatment (15 mg/kg, i.p., 5 times/week) groups. The tumor volume was measured on a weekly basis using a standard caliper. The tumor volume = 1/2(Length × Width^2^). The survival rate was recorded and plotted against time using GraphPad software. Mice were humanely euthanized, and the tumors were collected for further analyses. Animal studies were approved by the joint institutional research ethics review committee of the Tri-Service General Hospital and the National Defense Medical Center (approval number: LAC-2018-0291), and all experiments were consistent with those laid out in The National Academies of Science, Engineering, and Medicine Guide for the Care and Use of Laboratory Animals.

### 2.11. Immunohistochemistry

A standard immunohistochemical analysis (IHC) protocol was performed on the tumor sections collected from the in vivo experiment. Briefly, tumor sections (5 µm thick) were dewaxed by xylene (5 min, 2x) and rehydrated with ethanol gradient (100%, 95%, and 70%, each for 5 min), and followed by blocking of endogenous peroxidase activity using 3% hydrogen peroxide. Antigen retrieval process was carried using a microwave (power set at high), while the slides were immersed in ethylenediaminetetraacetic acid (10 mM EDTA, pH 8.0) for 2 min and blocked with 10% normal goat serum. The sections were then incubated with primary antibodies SREBP1 (ab28481; 1:100 dilution), ZEB1 (ab228986; 1:100 dilution), vimentin (ab92547; 1:100 dilution) from Abcam (Abcam, MA, USA), E-cadherin (20874-1-AP; 1:100) from Proteintech Group (Proteintech, IL, USA), and Ki-67 (MA5-14520, 1:100 dilution) from Thermo Fisher Scientific (Thermo Fisher Scientific, Waltham, MA, USA) overnight in cold, followed by the incubation with secondary antibody of goat anti-mouse IgG HRP-conjugated (1:10,000) using a HRP Polymer Kit (#TP-015-HD; Lab Vision, Fremont, CA, USA). The slides were then stained with diaminobenzidine (DAB) and counterstained with Gill’s hematoxylin (Thermo Fisher Scientific, Waltham, MA, USA).

### 2.12. Statistical Analyses

In vitro experiments were carried out independently 3 times. Data were analyzed using GraphPad Prism (La Jolla, CA, USA) and presented as the mean ± standard deviation (SD). The statistical difference between two groups were determined using Student’s t-test and one-way ANOVA was used for multiple group comparisons. A **p*-value ≤0.05 was considered statistically different.

## 3. Results

### 3.1. SREBP1 Expression Was Elevated in ESCC Tissues and Cell Lines

To determine whether SREBP1 is dysregulated in esophageal cancer, we first performed bioinformatics analysis using Oncomine to investigate the expression of SREBP1, and found that SREBP1 mRNA expression levels in ESCC tumors were higher than those in normal esophageal tissues in two independent datasets (Figure 1A) [28,29]. In addition, SREBP1 mRNA was estimated to be 2.95-fold higher in the ESCC samples as compared to the normal counterparts (Figure 1B) from the TCGA database (http://firebrowse.org/). Furthermore, an increased SREBP1 mRNA was significantly correlated to the shorter overall survival time of ESCC patients (Figure 1C) from the same TCGA ESCC cohort. Subsequently, we identified that SREBP1 is a target of tumor suppressor miR-142-5p (as one of the top-ranking miRs); miR-142-5p level was negatively associated with SREBP1 expression in ESCC patient cohort (*N* = 162, TCGA ESCA database) (Figure 1D). In connection, a higher level of miR-142 was found to be associated with a higher survival rate in an ESCC cohort (Figure 1E). Collectively, we hypothesized that increased SREBP1 and lowered miR-142-5p (targeting SREBP1) served as a poor biosignature for ESCC patients.

### 3.2. SREBP1 Expression Is Closely Associated with EMT and Metastatic Potential of ESCC

To functionally validate the role of SREBP1 in ESCC progression, we first knocked down its expression in one ESCC cell line, OE21, and overexpressed it in an esophageal adenocarcinoma cell (EAC) line, OE33. As our data show, SREPB1-silenced OE21 cells showed a significantly reduced colony-forming ability as compared to their control counterparts (Figure 2A), and the reversed observations were made in SREBP1-overexpressing OE33 cells (Figure 2A). Similarly, SREBP1-silenced OE21 cells exhibited a significantly reduced ability to migrate (Figure 2B) and invade (Figure 3C), while the increased migratory and invasive abilities were seen in the SREBP1-overexpressing OE33 cells. SREBP1 silencing was accompanied with reduced expression of ZEB1 and vimentin (Vim), while there was increased E-cadherin in OE21 cells (Figure 3D); the opposite phenomena were seen in the SREBP1-overexpressing OE33 cells (Figure 3D, also Appendix A). The observations strongly suggested that SREBP1 expression is positively associated with the mesenchymal status of the esophageal cancer cells.

### 3.3. Tumor Suppressor miR-142-5p Targets both SREBP1 and ZEB1

From our bioinformatics analysis of clinical cohorts of ESCC patients, a negative correlation exists between the level of tumor suppressor miR-142-5p and SREBP1. Here, we showed that SREBP1-silenced OE21 cells contained a significantly lower miR-142-5p as compared to the control counterpart (Figure 3A), whereas a significantly increased miR-142-5p was found in the SREBP1-overexpressing OE33 cells (Figure 3A). Using different target prediction software, we found that miR-142-5p could target both SREBP1 and ZEB1 (Figure 3B), as demonstrated by the potential 3′UTR binding of both genes with miR-142-5p. To test our hypothesis, we transfected miR-142-5p mimic (for overexpression of miR-142-5p) and inhibitor (to silence miR-142-5p) molecules into both cell lines. As expected, when miR-142-5p level was elevated by mimic molecules, a significantly lowered mRNA level (upper panels, Figure 3C) and protein level (lower panels, Figure 3C) of both REBP1 and ZEB1 were observed in both OE21 and OE33 cells, while the opposite trend was seen in the cells transfected with miR-142-5p inhibitor molecules. Consistently, the higher level of miR-142-5p by mimic molecules led to a significantly suppressed migratory ability of both OE21 and OE33 cells, whereas miR-142-5p-silenced OE21 and OE33 cells showed significantly increased migratory ability (Figure 3D). Notably, we also found that the self-renewal ability (by tumor sphere formation assay) was significantly suppressed in both cell lines transfected with miR-142-5p mimic molecules (Figure 3E). These observations provided the inverse relationship between the expression levels of miR-142-5p (tumor suppressor) and SREBP1 and established that miR-142-5p functioned to suppress esophageal tumorigenesis via the expression of both SREBP1 and ZEB1.

### 3.4. SREBP1 Inhibitor, Fatostatin, Suppressed ESCC Tumorigenesis, and Stemness

Inhibitors of SREBP1, such as fatostatin, have been previously shown to contain anticancer activity due to the fact that they negatively affect the lipid genesis of cancer cells [30,31]. However, this potential has not been fully explored in EC and cancer stem cells. Here, we showed that fatostatin treatment (72 h) effectively suppressed the cell viability in both parental and spheres of EC22 and EC33 cells (Figure 4A), where spheres were comparatively more resistant against the treatment. In addition, fatostatin treatment (10 μM, 24 h) suppressed the percentage of CD133+ population in both OE21 and OE33 spheres (Figure 4B). Fatostatin-mediated effects were shown by the Western blots of fatostatin-treated spheres of OE21 and OE33. Fatostatin treatment (10 μM, 48 h) resulted in the reduced expression of SREBP1, ZEB1, and vimentin, while it increased E-cadherin (Figure 4C) and increased miR-142-5p (Figure 4D).

### 3.5. In Vivo Validation of the Antineoplastic Function of Fatostatin

Finally, the anti-EC effects of fatostatin were evaluated using OE21 sphere-injected mice. Fatostatin-treated mice showed a significantly lower tumor burden over time as compared to the sham control (Figure 5A). More importantly, mice in the fatostatin group all survived after the eight week experiment, while only 40% in the vehicle control group survived (Figure 5B). The immunohistochemical analysis of the tumor sections showed that fatostatin-treated samples showed a markedly reduced staining for SREBP1, Ki67, ZEB1, and vimentin as compared to those in the control counterparts (Figure 5C). In addition, flow cytometric analysis of tumor cells harvested from the control and fatostatin mice indicated that fatostatin-treated tumor cells contained a lower percentage of CD133+ cells (Figure 5D) and higher miR-142-5p level (Figure 5E). Together, these observations provided support for our hypothesis, whereby inhibition of SREBP1 resulted in the suppression of ESCC tumorigenesis and stemness.

## 4. Discussion

SREBP1 is a critical transcription factor that controls the expression of genes important for the uptake and synthesis of lipids, such as cholesterol, fatty acids, and phospholipids [32]. Accumulating evidence suggests that SREBP1 facilitates tumor progression, and the upregulation of SREBP1 has often been detected in many cancer types [22,23,24,25,26,27,33,34,35,36]. In this study, we first identified an increased SREBP1 expression in different ESCC cohorts, which was in an inverse relationship with tumor suppressor miR-142-5p. Despite the previous studies on SREBP1, its tumorigenic roles have not been investigated fully in ESCC. We showed that the expression of SREBP1 was associated with the tumorigenic properties where it was associated with the EMT makers. Our results were supported by previous studies where increased SREBP1 expression facilitated EMT in breast and colon cancer [26,35]. In addition, studies have reported that SREBP1 is involved in the proliferation of multiple cancers [22,24,34,37]. In agreement, our gene-silencing echoed these reports, where SREBP1-silenced OE21 cells were significantly less capable of forming colonies as compared to their parental counterparts, while the opposite was observed in the SREBP1-overexpressing OE33 cells (Figure 1 and Figure 2).

Mechanistically, our data linked the oncogenic properties of SREBP1 to a concerted increase in ZEB1 and decreased miR-142-5p. Firstly, we identified both SREBP1 and ZEB1 as the silencing targets of miR-142-5p using multiple algorithms, and the negative correlation between their expression levels was found in the public databases of EC (Figure 3). Second, miR-142-5p has been demonstrated to serve as a tumor suppressor in different cancer types. For instance, a higher level of miR-142-5p-induced apoptosis was found in osteosarcoma via suppression of Erk-associated signaling [38]. In another example, miR-142-5p targeted several antiapoptotic genes, including baculoviral IAP repeat-containing 3 (BIRC3), B-cell lymphoma-2 (BCL2), BCL2-like 2 (BCL2L2), and myeloid cell leukemia sequence 1 (MCL1), and could improve cisplatin-mediated anticancer function in ovarian cancer [39]. Our observations, where miR-142-5p targeted both SREBP1 and ZEB1 in ESCC cells, provide additional support to its role as a tumor suppressor. More importantly, our results also provide a potential link between miR-142-5p and the self-renewal ability of ESCC cells where exogenous miR-142-5p led to a decreased ability to generate tumor spheres in both OE21 and OE33 cells. A recent study indicated that miR-142-5p functioned to suppress cell migration by targeting VCAM-1 in bone marrow-derived mesenchymal stem cells [40]. Trisaal et al. showed that loss-of-function the in miR-142 gene resulted in the increase in HOXA gene (a key gene for maintaining stemness in hematopoietic stem cells) and promoted leukemogenesis [41]. Together, the results from others and this study have provided evidence for the role of miR-142-5p as a tumor suppressor, while the decreased level of miR-142-5p was linked to the EMT and stemness of ESCC.

After establishing that SREBP1 elevation is associated with the EMT and malignant phenotypes in EC, we examined the therapeutic potential of targeting SREBP1 using fatostatin, a previously established SREBP inhibitor. Indeed, the fatostatin treatment led to decreased SREBP1 expression as well as vimentin, ZEB1, and metastatic potential of both OE21 and OE33 cancer cells. More importantly, tumor spheres generated from both OE21 and OE33 cells also responded to fatostatin treatment and significantly reduced the percentage of CD133+ cell populations within OE21 and OE33 cell lines (Figure 4). Our results were supported by previous studies, where fatostatin was shown to provide anticancer functions in different tumor types, such as pancreatic cancer and lung cancer [31,42]. More importantly, fatostatin was shown to disrupt estrogen-mediated signaling and suppress tumorigenesis in esophageal carcinoma [43,44,45], providing strong support to the current study. However, we believe this is the first report to demonstrate that fatostatin treatment inhibited the self-renewal ability of ESCC stem-like cells. We speculate that reduced self-renewal ability of ESCC spheres by fatostatin treatment could be associated with the reduced expression of ZEB1 and vimentin. Increased expression of mesenchymal markers, such as ZEB1, SNAIL, and Twist1, have been shown to induce cancer EMT and to be closely associated with the generation of cancer stem cells [46].

Previous reports have shown that fatostatin inhibits proliferation and induces apoptosis in ER+ breast cancer cells via the activation of endoplasmic reticulum (ER) stress and lipid accumulation, under lipid-sufficient conditions [47]; this could be attributed to the fact that fatostatin directly and negatively impacted on the lipid-synthesis and metabolic pathways [48]. This is supported by a previous study where ectopic expression of SREBP1 resulted in increased levels of lipogenic genes, such as fatty acid synthase (FASN), stearoyl CoA desaturase (SCD), and acetyl-CoA carboxylase-1 (ACC), and augmented lipogenesis and sphere formation in MCF10A stem-like cells [49]. In the same vein, the generation of tumor spheres is a process involving membrane remodeling and upregulated lipid metabolism [24], thus fatostatin-mediated suppression of SRBP1 (a key regulator of lipogenesis) led to the inhibition of tumor sphere generation. Our results from in vivo experiments using fatostatin also supported this hypothesis that fatostatin-treated mice showed a significantly lower tumor burden and better survival rate as compared to the vehicle control counterparts. Notably, tumor sections from fatostain-treated mice showed a significantly lower staining of SREPB1, ZEB1, and vimentin, while also showing increased E-cadherin, agreeing with the in vitro observations (Figure 5). Consistently, a significantly higher level of miR-142-5p was also detected in the tumor cells harvested from the fatostatin-treated group. Together, our in vitro and in vivo results lends support for further investigate the potential usage of fatostatin for the treatment of EC in the future.

It is important to note the limitations of the current study, where the role of estrogen receptor was not investigated. Al-Khyatt et al. [50] elegantly demonstrated that ERβ was the predominant form in both normal mucosa and esophageal cancer cells, whereas ERα was detected at a minimal level. In addition, these authors showed that the proliferation of OE33 and OE19 cell lines was does-dependently inhibited, while apoptosis was induced by an ERα-specific antagonist (MPP) and an ERβ-specific antagonist (PHTPP), establishing ESR1 and ESR2 as potential therapeutic targets for esophageal adenocarcinoma. A recent study established a model depicting that long-term aromatase inhibitors (AI) treatment promotes constitutive activation of SREBP1, which leads to reactivation of ERα and cytoskeletal rearrangements via Keratin-80 and promotes the invasive phenotype of breast cancer cells [27]. Together, observations from this study and others strongly suggest the therapeutic potential of agents such as fatostatin (this study) and 4-hydroxytamoxifen [51], which can both target estrogen receptors and disrupt lipid metabolism for the treatment of esophageal cancer.

In conclusion, our results graphic summary in Figure 6 demonstrates the upregulation of SREBP1 in OE21 tumors and cells and shows that its expression is correlated to the disease progression and poor prognosis. More importantly, we provided a link between increased tumorigenic properties in ESCC cells with an increased expression of SREBP1/ZEB1 and reduced miR-142-5p. Targeting the SREBP1/ZEB1/miR-142-5p signaling axis using fatostatin may represent an alternative and improved adjuvant option for treating malignant ESCC.

## Figures and Tables

**Figure 1 cells-09-00007-f001:**
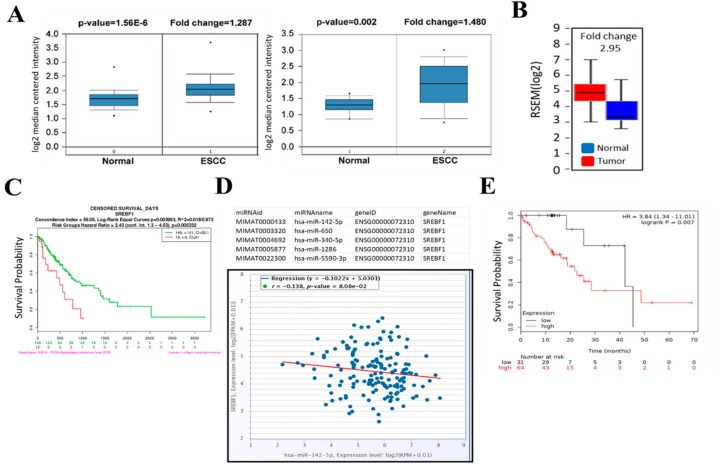
Increased SREBP1 (SREBF1) and decreased miR-142-5p expression is correlated with a poor clinical prognosis in patients with esophageal squamous cell carcinoma (ESCC). (**A**) Oncomine databases showed that SREBP1 expression was significantly higher in ESCC patients as compared to normal tissues. (**B**) TCGA ESCC cohort analysis showed that SREBP1 expression was approximately 2.95-fold higher in the ESCC tumors (*N* = 185) versus normal tissues (*N* = 11). (**C**) A higher SREBP1 mRNA was associated with a significantly shorter survival time (days) in the patients with ESCA (esophageal carcinoma, TCGA cohort). Log-rank *p* = 0.003993. (**D**) Target prediction analysis showed that miR-142-5p ranks as one of the top micorRNAs that targets SREBP1 (3 different algorithms were used for prediction); a negative correlation was identified between miR-142-5p and SREBP1 expression in patients with ESCC (*N* = 162), *p* = 8.08 × 10^−2^; (**E**) Kaplan–Meier survival curve shows that a higher level of miR-142-5p predicts a better survival probability in ESCC patients (*p* = 0.007).

**Figure 2 cells-09-00007-f002:**
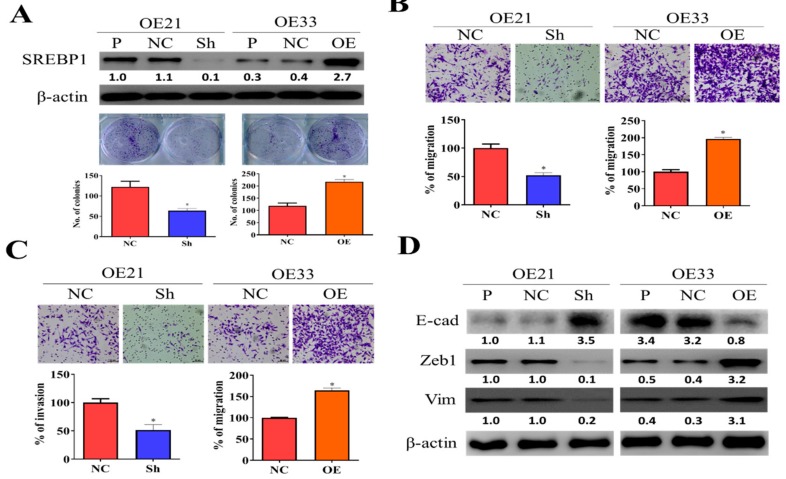
SREBP1 expression was positively correlated with the metastatic potential of ESCC cells. (**A**) Silencing SREBP1 in OE21 cells resulted in the significantly reduced colony-forming ability of OE21 cells (left panel), while overexpression of SREBP1 in OE33 cells showed the opposite effect (right panel). Migratory (**B**) and invasive (**C**) abilities were positively correlated to the expression of SREBP1 in ESCC cells. (**D**) Western blots of SREBP1-silenced OE21 and SREBP1-overexpressing OE33 cells. SREBP1-silenced OE21 cells showed a reduced expression of mesenchymal markers, ZEB1 and vimentin (vim), while increased epithelial marker, E-cadherin (E-cad); the opposite trend was observed in the SREBP1-overexpressing OE33 cells. **p* < 0.05.

**Figure 3 cells-09-00007-f003:**
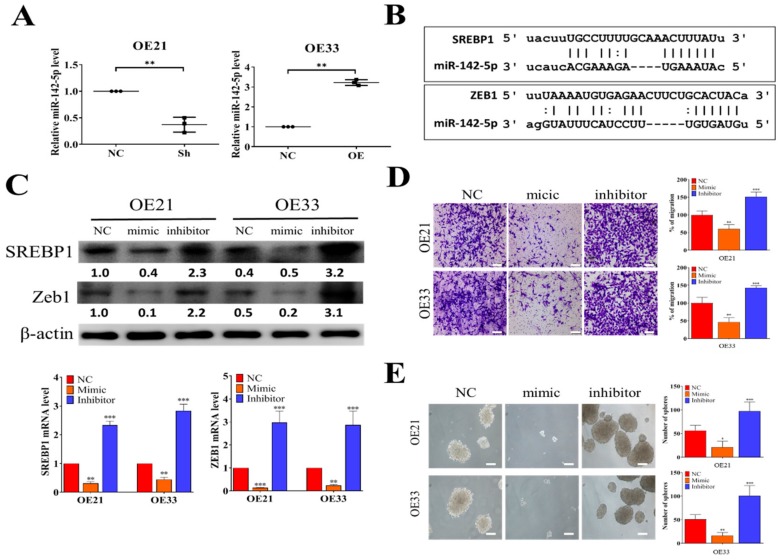
Tumor suppressor miR-142-5p targets SREPB1 and ZEB1 in ESCC. (**A**) QPCR analysis showed that SREBP1-silenced OE21 cells contained a significantly higher miR-142-5p level as compared to their negative control (NC) counterparts, while a lower miR-142-5p level was found in the SREPBP1-overexpressing OE33 cells. (**B**) Target prediction for miR-142-5p showed that SREBP1 and ZEB1 are both targets of miR-142-5p. The scheme shows the binding sequences of miR-142-5p to SREBP1 and ZEB1 at their 3′UTR (3′ untranslated region). (**C**) Comparative qPCR analyses and Western blots showed the reverse relationship between miR-142-5p and SREBP1 expression. Mic, mimic molecules of miR-142-5p; inh, inhibitor molecules of miR-142-5p; NC, negative control. (**D**) Transwell migratory assay indicated that when miR-142-5p was increased (by mic molecules), the migratory abilities of OE21 and OE33 were significantly reduced and the reverse was observed when miR-142-5p level was decreased by inhibitor molecules. (**E**) Tumor sphere formation assay results showed that increased miR-142-5p significantly reduced the tumor sphere-forming ability in both OE21 and OE33 cells, whereas an increased number of spheres were generated when miR-142-5p was suppressed (by inhibitor molecules). **p* < 0.05; ***p* < 0.01; ****p* < 0.001.

**Figure 4 cells-09-00007-f004:**
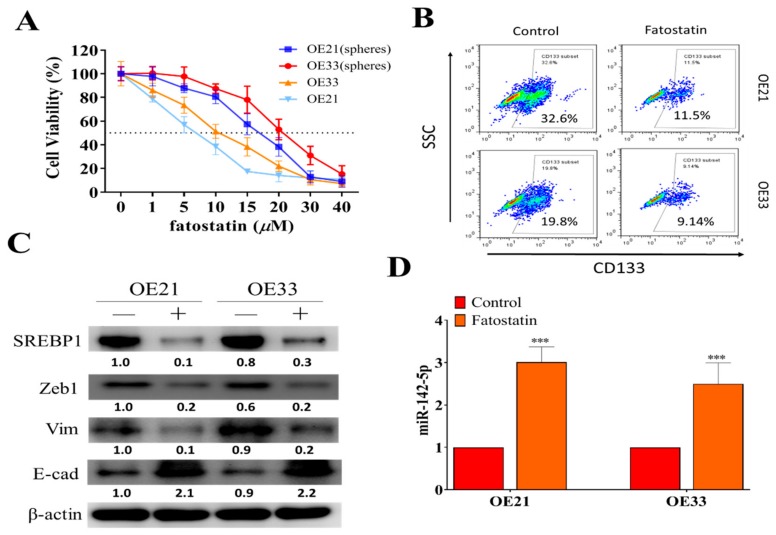
SREBP inhibitor, fatostatin, suppressed oncogenic phenotypes of ESCC. (**A**) Cell viability assay indicated both parental and spheres generated from OE21 and OE33 cells responded against fatostatin treatment. (**B**) Flow cytometric assay demonstrated that CD133+ cell percentages were reduced in both OE21 and OE33 spheres post-treatment of fatostatin (10 μM, 24 h). (**C**) Western blots of fatostatin-treated OE21 and OE33 cells showed that the treatment (lanes marked by +) reduced the expression of SREBP, ZEB1, and vimentin (Vim), while it increased E-cadherin (E-cad) as compared to the control (marked by -); GAPDH served as loading control. (**D**) qPCR analysis revealed that fatostatin treatment resulted in a significantly increased miR-142-5p level in both OE21 and OE3 cells. ****p* < 0.001.

**Figure 5 cells-09-00007-f005:**
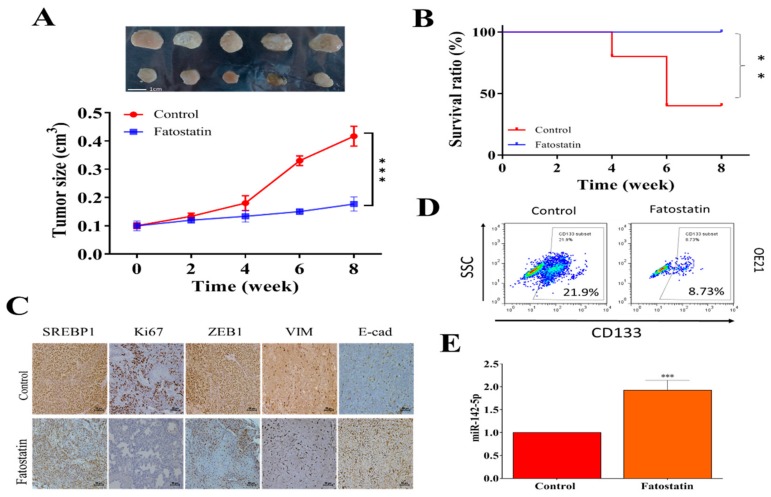
Evaluation of therapeutic function of fatostatin in an ESCC xenograft mouse model. (**A**) Tumor size versus time curve. Mice (injected with OE21, 2 × 10^6^ cells) that received fatostatin treatment (15 mg/kg, i.p., 5 times/week) showed a significantly smaller tumor size as compared to vehicle control counterparts. (**B**) Survival curve over time. After the 8-week experiment, the fatostatin treatment group showed 100% survival ratio, while 40% in the control group survived. (**C**) Immunohistochemical analysis of tumor sections shows decreased staining for SREPB1, Ki67, ZEB1, and Vim, while increased staining in E-cadherin was observed in fatostatin treatment samples as compared to their control counterparts. (**D**) Flow cytometry assay showed that tumor cells isolated from fatostatin-treated mice contained a lower percentage of CD133+ cells as compared to their control counterparts. (**E**) QPCR analysis indicated that tumor cells from fatostatin-treated mice expressed a significantly higher level of miR-142-5p. ***p* < 0.01; ****p* < 0.001.

**Figure 6 cells-09-00007-f006:**
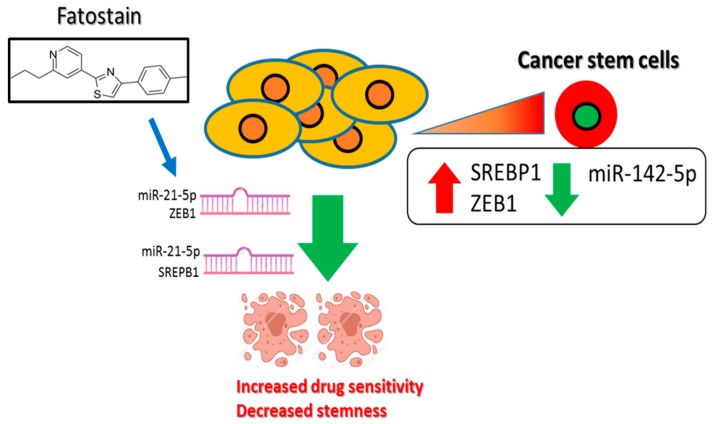
Increased SREBP1/ZEB1 and decreased miR-142-5p tumor suppressor level is associated with a poor prognosis of esophageal cancer and increased malignant properties, including stemness. Inhibition of SREBP1-associated signaling using fatostain is associated with the induction of miR-142-5p and reduced expression of its targets, ZEB1 and SREBP1, leading to the increased drug sensitivity and decreased stemness.

**Table 1 cells-09-00007-t001:** List of primer sequences for Real-Time PCR

Gene	Forward sequence	Reverse sequence
SREBP1	CGGCGCTGCTGACCGACATC	CCCTGCCCCACTCCCAGCAT
GAPDH	AGCCACATCGCTCAGACAC	GCCCAATACGACCAAATCC
miR-142-5p	AACTCCAGCTGGTCCTTAG	TCTTGAACCCTCATCCTGT
U6	GCTTCGGCAGCACATATACTAAAAT	CGCTTCACGAATTTGCGT

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
