# Peer review of "Disruption of Cancer Metabolic SREBP1/miR-142-5p Suppresses Epithelial–Mesenchymal Transition and Stemness in Esophageal Carcinoma"

_cells, 2019, doi:10.3390/cells9010007_

Round 1

Reviewer 1 Report

This study addressed role, function, and therapeutic potential of sterol regulatory element-binding protein 1 (SREBP1) in esophageal squamous cell carcinoma (ESCC). The paper is interesting and experiments very properly designed. However, there are several questions to address:

Why OE-33 line was chosen to investigate ESCC? OE-33 is esophageal adenocarcinoma cells, not ESCC. Fatostatin blocks estrogen receptor signaling as well. However, the role of estrogen signaling is was not addressed or discussed. Estrogen impacts E-cadherin expression and affects EMT. Why this point was ignored completely?

3. Did authors check the level of estrogen receptor (ERalpha and ERbeta) expression in used cell lines ? It is also necessary to test effects of other estrogen receptor inhibitors.

4. Effects of 4-OH-tamoxifen in OE-33 and OE-19 cells were shown previously. Line 39-40: “ We found that fatostatin (4-hydroxytamoxifen ) suppressed  the cell viability of OE21 and OE33 cells …” this information was already published. However, author did not cite previous publications and did not discuss their finding using previous information (Due et al., 2016; Sukocheva et al., 2013). Thus, discussion section requires serious improvement and extension of discussion considering the role of estrogen receptors in the observed effects.

Page 11. Line 8: “…fatostatin, a previously established SREBP inhibitor…” – and not only SREBP inhibitor. It is necessary to indicate here all other targets of 4-hydroxytamoxifen including estrogen receptors ( see review Katzenellenbogen BS, Choi I, Delage-Mourroux R, Ediger TR, Martini PG, Montano M, Sun J, Weis K, Katzenellenbogen JA. Molecular mechanisms of estrogen action: selective ligands and receptor pharmacology. J Steroid Biochem Mol Biol. 2000;74(5):279-85. Review).

Author Response

Answers to the comments:

Point-by-point responses to reviewer’s comments:

We would like to thank the reviewer for the thorough reading of our manuscript as well as their valuable comments. We have followed their comments closely and feel that they have further improved the readability and appeal of our work, as well as strengthened the manuscript. Below are our point-by-point responses.

Q1: Reviewer #1: This study addressed role, function, and therapeutic potential of sterol regulatory element-binding protein 1 (SREBP1) in esophageal squamous cell carcinoma (ESCC). The paper is interesting and experiments very properly designed. However, there are several questions to address.

A1: We thank the reviewer for the time taken to review our work, the critical assessment of our findings and the final satisfactory predisposition towards our revised work. The reviewer’s helpful suggestions has helped us improve the acceptability and appeal of our work.

Q2: Reviewer #1: Why OE-33 line was chosen to investigate ESCC? OE-33 is esophageal adenocarcinoma cells, not ESCC.

A2: We thank the reviewer for the correction. We apologize for this oversight. We have clarified this in the revised manuscript.

Q3: Reviewer #1: Fatostatin blocks estrogen receptor signaling as well. However, the role of estrogen signaling is was not addressed or discussed. Estrogen impacts E-cadherin expression and affects EMT. Why this point was ignored completely?

A3: We thank the reviewer for the insightful comments. We have included a more in-depth discussion in the discussion section. Please see our revised manuscript, page 11, lines 40~50.

It is important to note the limitations of the current study where the role of estrogen receptor was not investigated. Al-Khyatt et al.,[50] elegantly demonstrated that ERβ was the predominant form in both normal mucosa and esophageal cancer cells whereas ERα was detected at minimal level. In addition, these authors showed that the proliferation of OE33 and OE19 cell lines was does-dependently inhibited while apoptosis induced by ERα specific antagonist (MPP) and an ERβ specific antagonist (PHTPP), establishing ESR1 and ESR2 as potential therapeutic targets for esophageal adenocarcinoma.  A recent study established a model depicting that long-term aromatase inhibitors (AI) treatment promotes constitutive activation of SREBP1 which leads to reactivation of ERα and cytoskeletal re-arrangements, via Keratin-80 and promotes invasive phenotype of breast cancer cells[51]. Together, observations from this study and others strongly suggest the therapeutic potential of agents, fatostatin (this study) and 4-hydroxytamoxifen[52], which can both target estrogen receptors and disrupt lipid metabolism for the treatment of esophageal cancer.

The reference has been added in the page 18, line 7-18.

Al-Khyatt, W.; Tufarelli, C.; Khan, R.; Iftikhar, S.Y. Selective oestrogen receptor antagonists inhibit oesophageal cancer cell proliferation in vitro. BMC cancer 2018, 18, 121, doi:10.1186/s12885-018-4030-5. Perone, Y.; Farrugia, A.J.; Rodriguez-Meira, A.; Gyorffy, B.; Ion, C.; Uggetti, A.; Chronopoulos, A.; Marrazzo, P.; Faronato, M.; Shousha, S., et al. SREBP1 drives Keratin-80-dependent cytoskeletal changes and invasive behavior in endocrine-resistant ERalpha breast cancer. Nature communications 2019, 10, 2115, doi:10.1038/s41467-019-09676-y. Katzenellenbogen, B.S.; Choi, I.; Delage-Mourroux, R.; Ediger, T.R.; Martini, P.G.; Montano, M.; Sun, J.; Weis, K.; Katzenellenbogen, J.A. Molecular mechanisms of estrogen action: selective ligands and receptor pharmacology. The Journal of steroid biochemistry and molecular biology 2000, 74, 279-285, doi:10.1016/s0960-0760(00)00104-7.

Q4: Reviewer #1: Did authors check the level of estrogen receptor (ERalpha and ERbeta) expression in used cell lines ? It is also necessary to test effects of other estrogen receptor inhibitors.

A4: Reviewer’s suggestion was greatly appreciated. We did not examine the expression of ERα (ESR1) and ERβ (ESR2) in our study. However, Al-Khyatt et al.,[1] elegantly demonstrated that ERβ was the predominant form in both normal mucosa and esophageal cancer cells whereas ERα was detected at minimal level. In addition, these authors showed that the proliferation of OE33 and OE19 cell lines was does-dependently inhibited while apoptosis induced by ERα specific antagonist (MPP) and an ERβ specific antagonist (PHTPP), establishing ESR1 and ESR2 as potential therapeutic targets for esophageal adenocarcinoma. In addition, a recent study established a model depicting that long-term aromatase inhibitors (AI) treatment promotes constitutive activation of SREBP1 which leads to reactivation of ERα and cytoskeletal re-arrangements, via Keratin-80 and promotes invasive phenotype of breast cancer cells [2]. Collectively, these studies strongly suggest the therapeutic potential of agents which can target estrogen receptors and disrupt lipid metabolism for the treatment of esophageal cancer.

Al-Khyatt, W.; Tufarelli, C.; Khan, R.; Iftikhar, S.Y. Selective oestrogen receptor antagonists inhibit oesophageal cancer cell proliferation in vitro. BMC cancer 2018, 18, 121, doi:10.1186/s12885-018-4030-5. Perone, Y.; Farrugia, A.J.; Rodriguez-Meira, A.; Gyorffy, B.; Ion, C.; Uggetti, A.; Chronopoulos, A.; Marrazzo, P.; Faronato, M.; Shousha, S., et al. SREBP1 drives Keratin-80-dependent cytoskeletal changes and invasive behavior in endocrine-resistant ERalpha breast cancer. Nature communications 2019, 10, 2115, doi:10.1038/s41467-019-09676-y.

We now have included a more in-depth discussion in the discussion section. Please see our revised manuscript, page 11, lines 40~50.

It is important to note the limitations of the current study where the role of estrogen receptor was not investigated. Al-Khyatt et al.,[50] elegantly demonstrated that ERβ was the predominant form in both normal mucosa and esophageal cancer cells whereas ERα was detected at minimal level. In addition, these authors showed that the proliferation of OE33 and OE19 cell lines was does-dependently inhibited while apoptosis induced by ERα specific antagonist (MPP) and an ERβ specific antagonist (PHTPP), establishing ESR1 and ESR2 as potential therapeutic targets for esophageal adenocarcinoma.  A recent study established a model depicting that long-term aromatase inhibitors (AI) treatment promotes constitutive activation of SREBP1 which leads to reactivation of ERα and cytoskeletal re-arrangements, via Keratin-80 and promotes invasive phenotype of breast cancer cells[51]. Together, observations from this study and others strongly suggest the therapeutic potential of agents, fatostatin (this study) and 4-hydroxytamoxifen[52], which can both target estrogen receptors and disrupt lipid metabolism for the treatment of esophageal cancer.

The reference has been added in the page 18, line 7-18.

Al-Khyatt, W.; Tufarelli, C.; Khan, R.; Iftikhar, S.Y. Selective oestrogen receptor antagonists inhibit oesophageal cancer cell proliferation in vitro. BMC cancer 2018, 18, 121, doi:10.1186/s12885-018-4030-5. Perone, Y.; Farrugia, A.J.; Rodriguez-Meira, A.; Gyorffy, B.; Ion, C.; Uggetti, A.; Chronopoulos, A.; Marrazzo, P.; Faronato, M.; Shousha, S., et al. SREBP1 drives Keratin-80-dependent cytoskeletal changes and invasive behavior in endocrine-resistant ERalpha breast cancer. Nature communications 2019, 10, 2115, doi:10.1038/s41467-019-09676-y. Katzenellenbogen, B.S.; Choi, I.; Delage-Mourroux, R.; Ediger, T.R.; Martini, P.G.; Montano, M.; Sun, J.; Weis, K.; Katzenellenbogen, J.A. Molecular mechanisms of estrogen action: selective ligands and receptor pharmacology. The Journal of steroid biochemistry and molecular biology 2000, 74, 279-285, doi:10.1016/s0960-0760(00)00104-7.

Q5: Reviewer #1: Effects of 4-OH-tamoxifen in OE-33 and OE-19 cells were shown previously. Line 39-40: “ We found that fatostatin (4-hydroxytamoxifen ) suppressed  the cell viability of OE21 and OE33 cells …” this information was already published. However, author did not cite previous publications and did not discuss their finding using previous information (Due et al., 2016; Sukocheva et al., 2013). Thus, discussion section requires serious improvement and extension of discussion considering the role of estrogen receptors in the observed effects. Page 11. Line 8: “…fatostatin, a previously established SREBP inhibitor…” – and not only SREBP inhibitor. It is necessary to indicate here all other targets of 4-hydroxytamoxifen including estrogen receptors (see review Katzenellenbogen BS, Choi I, Delage-Mourroux R, Ediger TR, Martini PG, Montano M, Sun J, Weis K, Katzenellenbogen JA. Molecular mechanisms of estrogen action: selective ligands and receptor pharmacology. J Steroid Biochem Mol Biol. 2000;74(5):279-85. Review).

A5: We thank the reviewer for such an insightful comment. We apologize for our oversights. However, we did add another level of regulatory mechanism where miR-142-5p was found to be a suppressor of SREBP1 and esophageal carcinogenesis.  Reviewers constructive comments once again are extremely valuable. We have now included all the suggested references and included a more in-depth discussion on this issue in the discussion section (please see page 11, lines 13-15).

Our results were supported by previous studies where fatostatin was shown to provide anti-cancer functions in different tumor types such as pancreatic cancer and lung cancer [31,42]. More importantly, fatostatin was shown to disrupt estrogen-mediated signaling and suppress tumorigenesis in esophageal carcinoma [43-45], providing strong supports to current study.

Please also see our revised manuscript, page 11, lines 40~50.

It is important to note the limitations of the current study where the role of estrogen receptor was not investigated. Al-Khyatt et al.,[50] elegantly demonstrated that ERβ was the predominant form in both normal mucosa and esophageal cancer cells whereas ERα was detected at minimal level. In addition, these authors showed that the proliferation of OE33 and OE19 cell lines was does-dependently inhibited while apoptosis induced by ERα specific antagonist (MPP) and an ERβ specific antagonist (PHTPP), establishing ESR1 and ESR2 as potential therapeutic targets for esophageal adenocarcinoma.  A recent study established a model depicting that long-term aromatase inhibitors (AI) treatment promotes constitutive activation of SREBP1 which leads to reactivation of ERα and cytoskeletal re-arrangements, via Keratin-80 and promotes invasive phenotype of breast cancer cells[51]. Together, observations from this study and others strongly suggest the therapeutic potential of agents, fatostatin (this study) and 4-hydroxytamoxifen[52], which can both target estrogen receptors and disrupt lipid metabolism for the treatment of esophageal cancer.

The reference has been added in the page 18, line 7-18.

Al-Khyatt, W.; Tufarelli, C.; Khan, R.; Iftikhar, S.Y. Selective oestrogen receptor antagonists inhibit oesophageal cancer cell proliferation in vitro. BMC cancer 2018, 18, 121, doi:10.1186/s12885-018-4030-5. Perone, Y.; Farrugia, A.J.; Rodriguez-Meira, A.; Gyorffy, B.; Ion, C.; Uggetti, A.; Chronopoulos, A.; Marrazzo, P.; Faronato, M.; Shousha, S., et al. SREBP1 drives Keratin-80-dependent cytoskeletal changes and invasive behavior in endocrine-resistant ERalpha breast cancer. Nature communications 2019, 10, 2115, doi:10.1038/s41467-019-09676-y. Katzenellenbogen, B.S.; Choi, I.; Delage-Mourroux, R.; Ediger, T.R.; Martini, P.G.; Montano, M.; Sun, J.; Weis, K.; Katzenellenbogen, J.A. Molecular mechanisms of estrogen action: selective ligands and receptor pharmacology. The Journal of steroid biochemistry and molecular biology 2000, 74, 279-285, doi:10.1016/s0960-0760(00)00104-7.

Reviewer 2 Report

In this manuscript, the authors investigated the functional role of sterol regulatory element-binding protein 1 (SREBP1) in esophageal squamous cell carcinoma (ESCC). They observed that SREBP1 expression was elevated in ESCC tissues and cell lines and associated with EMT. By targeting SREBP, they observed a reduced tumor growth in vitro and in vivo. The overall data quality is good but with some weaknesses. These concerns should be addressed to strengthen further the manuscript.

"Disruption of cancer metabolism SREBP1/miR142-5p circuit..."  it should be: "Disruption of cancer metabolic SREBP1/miR 142-5p circuit..."

pag 2 lines 23-26. These articles should be cited: Nat Rev Mol Cell Biol. 2019 Feb;20(2):69-84. doi: 10.1038/s41580-018-0080-4. / Semin Cancer Biol. 2019 Oct;58:1-10. doi: 10.1016/j.semcancer.2018.11.004.

Antibodies code numbers should be provided.

Why SREBP1 expression has been knocked down in OE21 and why has been overexpressed in OE33 cells? Both cell lines are in E/M hybrid state, are there some functional differences that can explain for this decision? Please clarify in detail in the text.

pag 7 line 14. "The observations strongly suggested that SREBP1 expression is positively associated with the mesenchymal status of the ESCC cells". In my opinion, SREBP1 expression was associated with an increased expression of mesenchymal markers. In fact, the authors did not observe a complete E to M transition or viceversa. Please clarify.

The functional significance of SREBP1 overexpression is lacking. Can this induce a significant up-regulation of lipogenic enzymes and a metabolic reprogramming? This should be investigated and discussed.

pag 10 line 22. This should be considered: Biochim Biophys Acta Mol Cell Biol Lipids. 2019 Mar;1864(3):344-357. doi: 10.1016/j.bbalip.2018.12.011.

Author Response

Q1: Reviewer #2: In this manuscript, the authors investigated the functional role of sterol regulatory element-binding protein 1 (SREBP1) in esophageal squamous cell carcinoma (ESCC). They observed that SREBP1 expression was elevated in ESCC tissues and cell lines and associated with EMT. By targeting SREBP, they observed a reduced tumor growth in vitro and in vivo. The overall data quality is good but with some weaknesses. These concerns should be addressed to strengthen further the manuscript.

A1: We thank the reviewer for the time taken to review our work, the critical assessment of our findings and the final satisfactory predisposition towards our revised work. The reviewer’s helpful suggestions have helped us improve the acceptability and appeal of our work.

Q2: Reviewer #2: "Disruption of cancer metabolism SREBP1/miR142-5p circuit..."  it should be: "Disruption of cancer metabolic SREBP1/miR 142-5p circuit..."pag 2 lines 23-26. These articles should be cited: Nat Rev Mol Cell Biol. 2019 Feb;20(2):69-84. doi: 10.1038/s41580-018-0080-4. / Semin Cancer Biol. 2019 Oct;58:1-10. doi: 10.1016/j.semcancer.2018.11.004.

A2: We thank the reviewer for the above suggestions. We have revised the manuscript accordingly.

Q3: Reviewer #2: Antibodies code numbers should be provided.

A3: We thank the reviewer for the above suggestions. Antibodies code numbers should be provided in Material and Methods section and supplementary Table S1 (please see page 4, line 27-30 and page 5, line 38-46).  

Q4: Reviewer #2: Why SREBP1 expression has been knocked down in OE21 and why has been overexpressed in OE33 cells? Both cell lines are in E/M hybrid state, are there some functional differences that can explain for this decision? Please clarify in detail in the text.

A2: We thank the reviewer for the insightful comments. We also performed the silencing and overexpressing experiments in OE21 and OE33 cells and found the similar effects (please see our updated Supplementary Figure S1). Our rationale to use OE21 (an esophageal squamous cell carcinoma line, ESCC) and OE33 (an esophageal adenocarcinoma cell line, EAC) for test our hypothesis where SREBP1/miR-142-5p cascade plays an important role in different subtypes of esophageal cancer.

Please see the figure legend of Supplementary Figure S1 in page 13, line 7-13.

Supplementary Figure S1. Overexpression (OE) and silencing (Sh) of SREBP1 in OE21 and OE33 cells. (A) SREBP1-overexpressing OE21 showed increased ZEB1 expression accompanied with enhanced tumor sphere forming (upper panel) and invasive potential (lower panel). (B) SREBP1-silenced OE33 cells showed the opposite effects.

Q5: Reviewer #2: pag 7 line 14. "The observations strongly suggested that SREBP1 expression is positively associated with the mesenchymal status of the ESCC cells". In my opinion, SREBP1 expression was associated with an increased expression of mesenchymal markers. In fact, the authors did not observe a complete E to M transition or vice versa. Please clarify.

A2: Reviewer’s comment is greatly appreciated. As stated by the reviewer, SREBP1 expression was associated with an increased expression of mesenchymal markers but no a complete E to M transition or vice versa. This is an ongoing investigation in our laboratory, and we suspect the followings may contribute to this interesting observation. First, as evident by our western blots, the level of SREBP1 was silenced not completely abolished. The residual SREBP1 could still be functional thus resulting to decreased M markers but not completely; however, we did observe a significantly reduced invasive ability. We are currently attempting using CRSPR/Cas9 to completely knockout SREBP1 expression in vitro and hopefully will be able to obtain viable cells for further investigations.  Second, compensatory survival mechanisms could be triggered by other oncogenes post SREBP1-silencing. For instance, it was reported that SREBP1 interacts with c-MYC to enhance the binding of c-MYC to the promoter of the mesenchymal gene such as SNAIL [1]. We are also investigating further in order to better address this interesting issue.

Please also see the updated reference.

Zhai, D.; Cui, C.; Xie, L.; Cai, L.; Yu, J. Sterol regulatory element-binding protein 1 cooperates with c-Myc to promote epithelial-mesenchymal transition in colorectal cancer. Oncology letters 2018, 15, 5959-5965, doi:10.3892/ol.2018.8058.

Q6: Reviewer #2: The functional significance of SREBP1 overexpression is lacking. Can this induce a significant up-regulation of lipogenic enzymes and a metabolic reprogramming? This should be investigated and discussed.

A2: We thank the reviewer for the constructive advice. We observed overexpressing SREBP1 in esophageal cancer cells (please see our updated Supplementary Figure S1) resulted in the increased tumor sphere forming and invasive abilities while the silencing SREBP1 resulted in the opposite effects.

Please see the figure legend of Supplementary Figure S1 in page 13, line 7-13.

Supplementary Figure S1. Overexpression (OE) and silencing (Sh) of SREBP1 in OE21 and OE33 cells. (A) SREBP1-overexpressing OE21 showed increased ZEB1 expression accompanied with enhanced tumor sphere forming (upper panel) and invasive potential (lower panel). (B) SREBP1-silenced OE33 cells showed the opposite effects.

Please also see our revised manuscript, page 11, lines 22-31.

Previous reports have shown that fatostatin inhibits proliferation and induces apoptosis in ER+ breast cancer cells via the activation of endoplasmic reticulum (ER) stress and lipid accumulation, under lipid sufficient conditions [47];  this could be attributed to the fact that fatostatin directly and negatively impacted on the lipid-synthesis and metabolic pathways[48]. This is supported by a previous study where ectopic expression of SREBP1 resulted in increased levels of lipogenic genes such as faty acid shynthase (FASN), stearoyl CoA desaturase (SCD) and acetyl-CoA carboxylase-1 (ACC) and augmented lipogenesis and sphere formation in MCF10A stem-like cells [49]. In the same vein, the generation of tumor spheres is a process involving membrane remodeling and up-regulated lipid metabolism [24], thus fatostatin-mediated suppression of SRBP1 (a key regulator of lipogenesis) led to the inhibition of tumor sphere generation.

Q7: Reviewer #2: pag 10 line 22. This should be considered: Biochim Biophys Acta Mol Cell Biol Lipids. 2019 Mar;1864(3):344-357. doi: 10.1016/j.bbalip.2018.12.011.

A2: We thank the reviewer for the suggestion and this reference has been added into the discussion. 

The reference has been added in the page 16, line 43-44 to page 17, line 1-3.

Giudetti, A.M.; De Domenico, S.; Ragusa, A.; Lunetti, P.; Gaballo, A.; Franck, J.; Simeone, P.; Nicolardi, G.; De Nuccio, F.; Santino, A., et al. A specific lipid metabolic profile is associated with the epithelial mesenchymal transition program. Biochimica et biophysica acta. Molecular and cell biology of lipids 2019, 1864, 344-357, doi:10.1016/j.bbalip.2018.12.011.

Round 2

Reviewer 2 Report

The manuscript has been significantly improved and it is now suitable for the publication.